# Co-occurrence of non-communicable disease risk factors and its determinants among school-going adolescents of Kathmandu Metropolitan City

**Kalpana Tandon**[1], **Nabin Adhikari**[2], **Bikram Adhikari**[2], **Pranil Man Singh Pradhan**[3]*

1 Central Department of Public Health, Institute of Medicine, Tribhuvan University, Kathmandu, Nepal,
2 Department of Community Programs, Dhulikhel Hospital-Kathmandu University Hospital, Dhulikhel, Nepal,
3 Department of Community Medicine, Maharajgunj Medical Campus, Institute of Medicine, Tribhuvan University, Kathmandu, Nepal

* pranil.pradhan@gmail.com

**Data Availability Statement:** All relevant data are within the manuscript and its Supporting Information files.

**Funding:** The author(s) received no specific funding for this work.

## Abstract

### Background

Adolescence is a critical transition in human growth and adolescents tend to engage in various risky behaviors which are likely to continue into adulthood. Co-occurrence of non-communicable disease risk factors has the potential to increase risks of chronic disease comorbidity and increased mortality in later life. Behavioral risk factors are adopted due to changes in lifestyle and adolescents are more prone to acquire them. This study aimed to determine the prevalence and associated factors of co-occurrence of non-communicable disease risk factors among school-going adolescents of Kathmandu Metropolitan City.

### Methods

We conducted a cross-sectional study among school-going adolescents of Kathmandu Metropolitan City in January/February 2020. We used stratified random sampling to select 1108 adolescents studying in 9, 10, 11, and 12 grades. We used Global Schools Health Survey tools to collect data. We entered data in EpiData 3.1 and exported it into Statistical Package for Social Science (SPSS) version 20 for statistical analysis. We estimated prevalence of NCDs risk factors and co-occurrence of risk factors. We applied multivariate multinomial logistic regression analysis adjusting for age, gender, ethnicity, religion, education, type of school, and parental education to determine factors associated with co-occurrence of NCDs risk factors.

### Results

The prevalence of physical inactivity, unhealthy diet, harmful use of alcohol and tobacco among school-going adolescents were 72.3% (95%CI: 69.6–74.9), 41.1% (95%CI: 38.2–44.0), 14.8% (95%CI: 12.8–17.0) and 7.8% (95%CI:6.3–9.5) respectively. The adolescent with co-occurrence of two or more risk factors was 40.7% (95%CI: 37.8–43.7). The school-

**Competing interests:** The authors have declared that no competing interests exist.

going adolescents who were in higher age group (AOR = 1.72, 95% CI- 1.06, 2.77), Hindus (AOR = 1.78, 95% CI-1.09, 2.89), other than Brahmin/Chhetri by ethnicity (AOR = 2.11, 95% CI-1.39, 2.22) and with lower education level of mothers (AOR = 2.40, 95% CI-1.46,3.98) were more likely to have co-occurrence of NCDs risk factors after adjusting for all socio-demographic variables.

## Conclusion

The co-occurrence of non-communicable disease risk factors was high among school going adolescents and was associated with age, religion, ethnicity and mother's education. Integrated and comprehensive interventional programs should be developed by concerned authorities.

## Introduction

Non-communicable diseases (NCDs) are slow and progressive with cardiovascular disease, cancer, chronic respiratory disease, and diabetes being the major NCDs [1, 2]. Globally these NCDs were the leading causes of mortality contributing to 71% (41million) of the 57 million deaths in 2016 [3, 4]. About 78% of all NCDs deaths and 85% of premature deaths occur in low and middle-income countries (LMIC) [1].

In Nepal, NCDs are a leading cause of deaths contributing to 66% of all deaths in 2017. Similarly, 49% of years of life lost (YLLs) due to premature death, and 59% of disability-adjusted life years lost (DALYs) are due to NCDs [5]. According to the Nepal NCDs risk factor survey 2013, 19% were current smokers, 17% were alcohol users almost 2% had low physical activity and unhealthy diet, 11% consumed processed food and 99% did not consume adequate vegetables and fruits as recommended by World Health Organization [6].

Some studies showed that about 70% of premature death in adults is due to lifestyle health-risk behaviors established during the adolescence period. NCDs such as diabetes, obesity, and cardiovascular diseases are preventable and can be delayed by addressing the risk factors at an earlier stage of adolescence [7].

The international report with the analysis of the distribution of health behaviors among adolescents from more than hundred countries by World Health Organization (2005/2006) found that approximately 80% of adolescents performed daily physical activities for at least 60 min, 25% had an unhealthy diet, 7.6% consumed beer weekly, and 6% smoked cigarettes daily [8]. Each of the risk factors impacts the health of the individual. Adolescents with low levels of physical activity and sedentary life have a higher risk of hypertension and obesity along with different causes and consequences affecting health [9]. Unhealthy diet, either low consumption of fruits and vegetables or the high intake of sugar rich food increase risk of obesity and diabetes mellitus type II [10]. Excessive alcohol use can alter the central nervous system functioning which may lead to depression and suicide among adolescents [11].

Risk factor studies among school-going adolescents in LMIC have shown that 14.9% had only one risk factor whereas one-third of participants had an occurrence of two or more risk factors [12]. The high prevalence of four common risk factors had been associated with a rise in NCD mortality by 3.35 fold [13]. Approximately, 34% of the total disease burden (DALYs) can be prevented by preventing these modifiable risk factors [14]. Schools can play a significant role in promoting a healthy lifestyle if the pattern of co-occurrence of NCD risk factors is

known among students [15]. Integrated, comprehensive, and coordinated approaches could strengthen and respond to multiple conditions or risk factors.

Co-occurrence has the potential to increase risks of chronic disease co-morbidity and increased mortality in later life [16]. To our knowledge, very few research were carried out in co-occurrence of the risk behavior in both adolescent and adult populations globally. Adolescents, who comprise almost 24% of the total population in the context of Nepal, are neglected in NCDs-related studies. Policies and programs targeting adolescents in maintaining and establishing healthy behavior are easier and cost-effective than changing unhealthy risk behavior at the later stage of life. We aimed to determine the pattern of co-occurrence of NCDs risk factors and its associated factors among school-going adolescents which will help in planning and implementation of integrated programs at the school and national level.

## Materials and methods

### Study design and population

We conducted a cross-sectional study among school-going adolescent students aged 13 to 19 years studying in grades 9 to12 in public and private schools of Kathmandu Metropolitan City.

### Participant recruitment

We calculated sample size to be 1108 using Cochrane formula assuming 11.2% prevalence of NCDs clustering at least three risk factors [5], 95% confidence interval, 3% absolute error, design effect of 2, and 31% non-response rate. We selected 10 schools (4 public and 6 private) out of 603 schools randomly. The public and private schools were assumed as strata and students were selected by population proportionate to the size sampling technique. The public and private schools held 66% and 34% of total students respectively. Thus, we selected 734 students from public and 374 from private schools for the study. From each selected school, all students present in classes 9, 10, 11, and 12 (if multiple sections were present in a class, one section was randomly selected) were included in the study.

We recruited respondents who met the eligibility criteria. The inclusion criteria included a) school going adolescents from grade 9 to 12 b) aged 13 to 19 years. Exclusion criteria included a) those who were not present on the day of data collection b) those whose parents did not give assent.

### Study settings

We conducted this study in public and private schools of Kathmandu Metropolitan City. Kathmandu is the capital of the Federal Democratic Republic of Nepal located in Province number three. It is also the largest metropolitan city in Nepal.

### Variables and measure

**Co-occurrence of NCDs risk factors.**   The dependent variable in the study was the co-occurrence of NCD risk factors [17]. It refers to the existence of two or more (out of the four) NCDs risk factors in an individual at the time of the data collection. The risk factors included: alcohol consumption, tobacco use, unhealthy diet, and physical inactivity. ***Alcohol consumption*** was defined as the current use of alcohol (at least one drink of alcohol on at least one day during the 30 days before the study. The intake of approximately 60 gm of alcohol results in a blood concentration of 0.8g/dl. A dose of alcohol was considered as a can of beer, a glass of wine, and a shot of whisky, rum vodka, or other. ***Tobacco use*** was defined as currently using any tobacco product (used any tobacco product on at least 1 day during 30 days before the

study). ***Unhealthy diet*** was defined as having any one of the following: Eating fruits and vegetables less than 5 times a day, eating from fast-food restaurants one or more days, and drinking carbonated soft drinks one or more times per day during the past 7 days. ***Sufficient physical activity*** was defined as being physically active at least 60 minutes per day during the 7 days before the study considering any type of physical activity that increased the heart rate and breathing of adolescents. Adolescents that practiced physical activities less than five times weekly were considered physically inactive. This parameter was based on evidence demonstrating that it is necessary to practice 60 minutes of physical activity five days a week for health maintenance during adolescence.

**Socio-demographic characteristics.** It included age (in years), sex (male/female), ethnicity (Brahmin/ Non Brahmins), types of school (public/private), level of education (secondary/higher secondary), and parents' education level. The ethnicity of the participants was classified as Dalit, Madhesi, Muslim, Brahmin/ Chhetri, and Other (Thakuri and Dasnamis). The level of education was categorized as secondary (grade 9 and 10) and higher secondary (11 and 12 grade).

## Data collection tools and techniques

We used a validated structured questionnaire developed by the Global School Health Survey 2015, World Health Organization (34). We collected data in January/February 2020. Permission was taken from the concerned school's authority. Sections were selected randomly. After selecting the class, all students were provided with information about the objectives of the study. On the first day, written informed consent forms were given to parents or guardians through class teachers and who returned the form the following day. On the following day, questionnaires were administered to those students who provided written consent from their parents/guardians. Students were explained the importance of responding to each question. Information regarding the identity of the students was not collected to maintain confidentiality. The seating arrangement was done with enough spacing to avoid contamination of responses. Students were informed about their voluntary participation. School authorities and teachers were not allowed to stay in the classroom while students were filling the questionnaires. Students were given 25–30 minutes to complete the form. The filled questionnaire was checked for completeness.

## Data quality control

Content validity was ensured by the guidance of experts and extensive literature throughout the preparation of research proposals. The tools used were based on the Global School Health Survey (GSHS) 2015, Nepal which was already tested and validated in the Nepalese context. The questionnaire was developed based on study objectives through extensive literature review using related articles with necessary modifications. The tool was translated into Nepali and reverse translated into the English language. The principal researcher herself was involved in data collection, analysis, and interpretation. To ensure reliability, pre-testing of the tool was done in 10 percent of non-sampled schools in Kathmandu. All necessary modifications were made after consultation with the supervisor and experts.

## Statistical analysis

The collected data were systematically coded and entered into EpiData Entry 3.1. The entered data were exported into IBM SPSS Version 20 where consistency was checked and cleaning and editing of data were done.

We performed descriptive analysis and presented parametric numerical variables with mean and standard deviation and categorical variables with percentage and frequency. We estimated

the prevalence of risk factors and co-occurrence of risk factors for NCDs among adolescents. We used the Clopper-Pearson and Goodman methods to determine the confidence interval of proportion. We applied multivariate multinomial logistic regression to determine factors associated with co-occurrence of NCDs risk factors. In the model we added socio-demographic variables- age, gender, ethnicity, religion, education, type of school, and parental education (mother's education, and father's education) with p-value less than 0.50 in univariate analysis (S1 Table). We estimated the adjusted odds ratio along with a 95% confidence interval.

### Ethical consideration

We obtained ethical clearance and approval from the Institutional Review Committee (IRC) of the Institute of Medicine (IOM), Maharajgunj (Reference number: 225-076/077, 27th November 2019). We obtained permission from the Education Section of Kathmandu Metropolitan City and the concerned school authorities. We obtained written informed consent from parents and students before including them in the study. We ensured voluntary participation and the freedom to refuse by the participants at any time. We maintained the confidentiality and anonymity of the students throughout the study. There was no risk to participants during data collection.

## Results

### Socio-demographic characteristics of school going adolescents

Out of 1108 respondents, 54.5% were males. The age of the participants ranged from 10 to 19 years with a mean age of 16.07 ±1.77 years. Of the total participants, the majority were Hindus (79.7%). Most of the participants were Janajati (46.5%) and Brahmin/Chhetri (46.3%) followed by Dalit and Madhesi ethnic groups. More than half of respondents had at least three close friends with whom they shared their feelings and were affected by their behavior (Table 1).

### Distribution of NCDs risk factors among school going adolescents

Table 2 presents the distribution NCDs risk factors among school going adolescents. Most of the participants did not consume vegetables/fruits more than five times a day. Less than half of the participants did not drink carbonated products less than five times and 23.2% of participants did not eat fast food in the past week. Similarly, two-fifths of the participants did not watch any advertisement related to the promotion of carbonated drinks/fast food on television. One-seventh of participants had ever smoking. Among them most smokers had initiated smoking at the age of 15 to 19 years of age (53.3%) followed by between 10 to 14 years of age (31.6%). The prevalence of current smoking was 6.8%. Only 2.9% of participants use tobacco other than cigarettes. One-fourth of the participants' parents were also using some form of tobacco. Less than one-third (30.7%) of participants had ever take alcohol. Among them half (50.3%) of the participants had the first drink of alcohol at the age of 15 to 19 years. One-seventh (14.8%) of the participants drank at least one drink during the last thirty days. Most of them (34.7%), obtain the alcohol from the family. Most of the parents of the participants were aware of the drinking habits of the children (53.3%). More than three-fourth (76.4%) of participants reported that they were taught about the harmful effects of alcohol in the schools. A majority (72.3%) of participants were physically inactive.

### Prevalence of NCDs risk factors among school going adolescents

Fig 1 presents the prevalence of NCDs risk factors among school going adolescents. The prevalence of physical inactivity, unhealthy diet (carbonated drinks, fast food, and not eating fruits

**Table 1. Socio-demographic characteristics of the school going adolescents (n = 1108).**

| Characteristics | n (%) | | |
|---|---|---|---|
| | Overall | Public | Private |
| **Age group (years)** | | | |
| 13–16 | 653(58.9 | 370(50.6) | 283(75.1) |
| 17–19 | 455 (41.1) | 361(49.4) | 94(24.9) |
| **Gender** | | | |
| Male | 604 (54.5) | 385(52.7) | 219(58.1) |
| Female | 504 (45.5) | 346(47.3) | 158(41.9) |
| **Grade** | | | |
| Secondary level | 635 (57.3) | 356(48.7) | 279(74.0) |
| Higher Secondary | 473 (42.7) | 375(51.6) | 98(26.0) |
| **Ethnicity** | | | |
| Brahmin/Chhetri | 513 (46.3) | 307(42.0) | 206(54.6) |
| Janajati | 515 (46.5) | 364(49.8) | 151(40.1) |
| Dalit | 59 (5.3) | 46(6.3) | 13(3.4) |
| Madhesi | 21 (1.9) | 14(1.9) | 7(1.9) |
| **Religion** | | | |
| Hinduism | 883 (79.7) | 563(77.0) | 320(84.9) |
| Buddhism | 165 (14.9) | 123(16.8) | 42(11.1) |
| Christianity | 42 (3.8) | 34(4.7) | 8(2.1) |
| Islam | 18 (1.6) | 11(1.5) | 7(1.9) |
| **Currently living with** | | | |
| Both Parents | 697 (62.9) | 417(57.0) | 280(74.3) |
| Others than Parents | 290 (26.2) | 223(30.5) | 67(17.8) |
| Single Parents | 121 (10.9) | 91(12.4) | 30(8.0) |
| **Number of close friends** | | | |
| At least 3 | 617(55.7) | 388(53.1) | 229(60.7) |
| More Less than three | 491(44.3) | 343(46.9) | 148(39.3) |
| **Father's education** | | | |
| Secondary level and below | 747 (67.4) | 547(74.8) | 200(53.1) |
| Higher secondary level & above | 361 (32.4) | 184(25.2) | 177(46.9) |
| **Mother's education** | | | |
| Secondary level and below | 860 (77.6) | 628(85.9) | 232(61.5) |
| Higher secondary level & above | 248 (22.4) | 103(14.1) | 145(38.5) |
| **Father's occupation** | | | |
| Household chores | 63 (5.7) | 52(6.1) | 11(2.9) |
| Business | 379 (34.2) | 225(30.8) | 154(40.8) |
| Services | 274 (24.7) | 152(20.8) | 122(32.4) |
| Agriculture | 169 (15.3) | 139(19.0) | 30(8.0) |
| Abroad | 141 (12.3) | 95(13.0) | 46(12.2) |
| Others | 82 (7.4) | 68(9.3) | 14(3.7) |
| **Mother's occupation** | | | |
| Household chores | 660 (59.6) | 450(61.6) | 210(55.7) |
| Business | 182 (16.4) | 101(13.8) | 81(21.5) |
| Services | 82 (7.4) | 32(4.4) | 50(13.5) |
| Agriculture | 118 (10.6) | 98(13.4) | 20(5.3) |
| Abroad | 25 (2.3) | 20(2.7) | 5(1.3) |
| Others | 41 (3.7) | 30(4.1) | 11(2.9) |

**Table 2. Distribution of NCDs risk factors among school going adolescents.**

| Risk factors related information (N = 1108) | n (%) | | |
|---|---|---|---|
| | Overall | Public | private |
| **Eating fruits in last 7 days** | | | |
| None | 215 (19.4) | 137(18.7) | 78(20.7) |
| Less than 5 times | 845 (76.3) | 561(76.7) | 284(75.3) |
| More than or equal to 5 times | 48 (4.3) | 33(4.5) | 15(4.0) |
| **Eating vegetable** | | | |
| None | 55 (5.0) | 41(5.6) | 14(3.7) |
| Less than 5 times | 982 (88.7) | 654(89.5) | 328(87.0) |
| More than or equal to 5 times | 71 (6.4) | 36(4.9) | 34(9.0) |
| **Carbonated drinks** | | | |
| None | 518 (46.8) | 336(46) | 182(48.3) |
| Less than 5 times | 563 (50.9) | 377(51.5) | 186(49.3) |
| More than or equal to 5 times | 27 (2.4) | 18(2.5) | 9(2.4) |
| **Fast food** | | | |
| Not eat | 257 (23.2) | 180(24.6) | 77(20.4) |
| Less than 5 times | 759 (68.6) | 500(68.4) | 259(68.7) |
| More than or equal to 5 times | 92 (8.2) | 51(7) | 41(10.9) |
| **Advertisement related to carbonated drinks/fast food** | | | |
| Not watched | 468 (42.2) | 318(43.5) | 150(39.8) |
| Watched a lot | 328 (29.6) | 229(31.3) | 99(26.3) |
| Very few/ less time watched | 311 (28.1) | 183(25.0) | 128(34.0) |
| **Taught about benefits of fruits and vegetables** | | | |
| No | 296 (26.7) | 179(24.5) | 117(31.0) |
| Yes | 812 (73.3) | 552(75.5) | 260(69.0) |
| **Tobacco-related responses** | | | |
| **Ever smoking** | | | |
| Yes | 155(14) | 110(15) | 45(11.9) |
| No | 953(86) | 621(85) | 332(88.1) |
| **Age(years) of initiation smoking** | | | |
| ≤ 9 | 23(14.8) | 16(14.5) | 7(16) |
| 10–14 | 49(31.6) | 36(32.7) | 13(28.5) |
| 15–19 | 83(53.5) | 58(52.8) | 25(5.5) |
| **Current smoker** | | | |
| No | 1033 (93.2) | 683(93.4) | 350(92.8) |
| Yes | 75 (6.8) | 48(6.4) | 27(7.2) |
| **Tobacco users other than cigarettes** | | | |
| No | 1076 (97.1) | 731(97.5) | 363(96.3) |
| Yes | 32 (2.9) | 18(2.5) | 14(3.7) |
| **Parents taking any form of tobacco** | | | |
| Neither of them | 783 (70.7) | 527(72.1) | 256(67.9) |
| Either father or mothers | 279 (25.2) | 165(22.6) | 114(30.2) |
| Both parents are users | 46 (4.2) | 39(5.3) | 7(1.9) |
| **Alcohol-related response** | | | |
| Yes | 340(30.7) | 203(27.8) | 137(36.3) |
| No | 768(69.3) | 528(72.2) | 240(63.7) |
| **Age (years) at first drink of alcohol other than few sips** | | | |

*(Continued)*

**Table 2.** (Continued)

| Risk factors related information (N = 1108) | n (%) | | |
|---|---|---|---|
| | **Overall** | **Public** | **private** |
| ≤ 9 | 66(19.4) | 30(14.7) | 36(26.2) |
| 10–14 | 103 (30.3) | 63(31) | 40(29.1) |
| 15–19 | 171 (50.3) | 110(54.3) | 61(44.5) |
| **Ever drink** | | | |
| No | 768(69.3) | 528(72.2) | 240(63.7) |
| Yes | 340(31.7) | 203(27.8) | 137(36.3) |
| **At least one alcoholic drink during the last 30 days** | | | |
| 0 days | 944 (85.2) | 635(86.9) | 309(82) |
| 1–2 days | 136 (12.3) | 85(11.6) | 51(13.5) |
| ≥ 3 days | 28 (2.5) | 11(1.5) | 17(4.5) |
| **Place where alcohol was obtained** | | | |
| Store | 48 (29.2) | 32(35.5) | 16(21.7) |
| From family | 57 (34.7) | 14(15.5) | 43(58.1) |
| From friends | 31 (18.9) | 20(22.2) | 11(14.8) |
| Gave money someone to brought | 20 (12.1) | 18 (20) | 2(2.7) |
| Stole | 8 (4.8) | 6(6.8) | 2(2.7) |
| Store | 48 (29.2) | 32(35.5) | 16(21.7) |
| From family | 57 (34.7) | 14(15.5) | 43(58.1) |
| **Did parents know that you drink alcohol?** | | | |
| No | 159 (46.7) | 112(51.3) | 47(38.5) |
| Yes | 181 (53.3) | 106(48.7) | 75(61.5) |
| **Taught about the problems associated with alcohol** | | | |
| No | 308 (27.8) | 227(31.1) | 81(21.5) |
| Yes | 800 (72.2) | 504(68.9) | 296(78.5) |
| **Physical activity related response (N = 1108)** | | | |
| **No of days active (at least 60min/day in last 7 days)** | | | |
| < 5 days | 801 (72.3) | 538(73.5) | 263(6.9) |
| ≥ 5 days | 307 (27.7) | 193(26.5) | 114(30.2) |
| **Having physical classes** | | | |
| No | 522 (47.1) | 317(43.4) | 205(54.4) |
| Yes | 586 (52.9) | 414(56.6) | 172(45.6) |
| **Taught about benefit of physical activity** | | | |
| No | 262 (23.6) | 165(22.6) | 97(25.7) |
| Yes | 846 (76.4) | 566(77.4) | 280(74.3) |
| **Usual spending time on sitting, watching TV, playing computer when you are not in school** | | | |
| <1 hours | 314 (28.3) | 239(32.7) | 75(19.9) |
| 1–4 hours | 555 (50.1) | 345(47.2) | 210(52.8) |
| ≥ 5 hours | 239(21.6) | 147(20.1) | 92(24.4) |

and vegetables), alcohol intake, and tobacco were 72.3(95% CI: 69.6–74.9), 41.1(95% CI: 38.2–44.0), 14.8(95% CI: 12.8–17.0), and 7.8(95% CI: 6.3–9.5), respectively. The prevalence of tobacco use was higher in private school (8.5% (95% CI: 5.9–11.8)) compared to public school (7.4% (95% CI: 5.6–9.5%). Similarly, alcohol use was also higher in private school (18.0% (95% CI: 14.3–22.3) compared to public school (13.1% (95%CI: 10.8–15.8). (Fig 1).

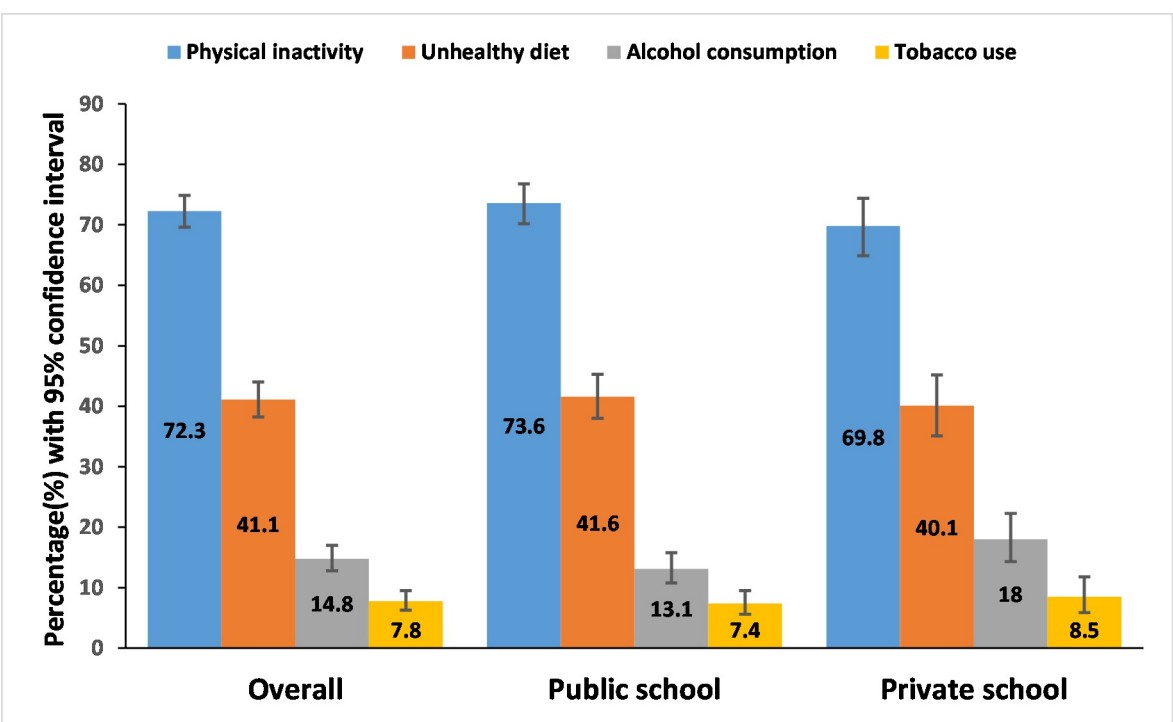

**Fig 1. Prevalence of NCDs risk factors among school going adolescents.**

### Prevalence of co-occurrence of risk factors among school going adolescents

Fig 2 presents the prevalence of co-occurrence of risk factors among school going adolescents. Among total respondents, only 14.3% (95% CI: 12.3–16.5) were not having any NCDs risk factors. The prevalence of co-occurrence of NCDs risk factors was 40.7% (95%CI: 37.8–43.7) accounting 40.1% (95% CI: 35.1–45.2) among students of private school and 41.0% (95%CI: 37.4–44.7%) among students of public school. A majority, 45.0% (95%CI: 42.1–48.0), of the respondents had one out of four risk behavior whereas 1.6% (95%CI: 1.0–2.6) of participants had all four risk factors (Fig 2).

### Association of socio-demographic variables with co-occurrence of NCDs risk factors

Table 3 shows the multivariate analysis from multinomial logistic regression model to determine the independent association between the independent and dependent variables. From multinomial regression, it was found that the age of participants, ethnicity, and mothers' education was significantly associated with the co-occurrence of NCDs risk factors.

The odds of co-occurrence of risk factors was 1.72 (95%CI:1.06, 2.77; p-value:0.027) times more among 17–20 years old respondents compared to 13–17 years old respondents after adjusting for all socio-demographic variables. The odds of co-occurrence of risk factors was 1.78 (95% CI 1.09–2.89, p-value 0.020) times among Hindus compared to Non-Hindus. Non-Brahmin/Chhetris were 2.11(95% CI 1.39–2.22, p-value <0.001) times less likely to have co-occurrence of risk factors compared to Brahmin/Chhetris after adjusting for all socio-demographic variables. The odds of co-occurrence of risk factors was 2.40(95% CI 1.46–3.98, p-value <0.001) times among participants whose mother education was secondary and below after adjusting for all socio-demographic variables.

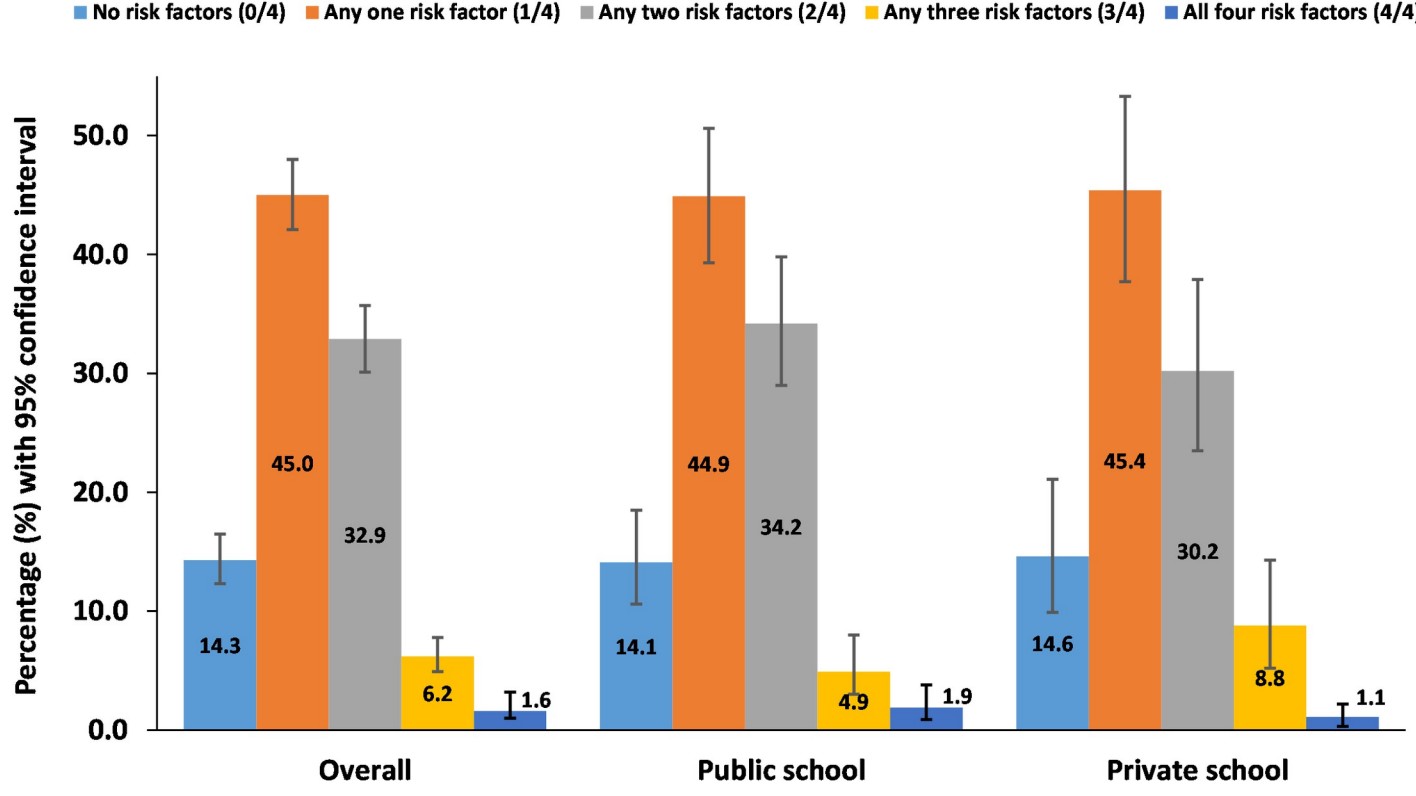

**Fig 2. Prevalence of co-occurrence of risk factors among school going adolescents.**

## Discussion

Our results showed the prevalence of physical inactivity, unhealthy diet, alcohol intake, and tobacco were 72%, 41%, 15%, and 8% respectively. Forty-five percent of the respondents had

**Table 3. Association of socio-demographic variables with co-occurrence of NCDs risk factors.**

| Variables | Categories | One risk factor versus No risk factors | | Co-occurrence versus No risk factors | |
|---|---|---|---|---|---|
| | | Adjusted odds ratio (AOR) | p-value | Adjusted odds ratio (AOR) | p-value |
| Gender | Male | 1 | | 1 | |
| | Female | 1.42(0.98, 2.05) | 0.064 | 1.31(0.89 1.91) | 0.161 |
| Age | 13–16 | 1 | | 1 | |
| | 17–19 | 1.30(0.81, 2.09) | 0.274 | 1.72(1.06, 2.77) | 0.027 |
| Religion | Non-Hindu | 1 | | 1 | |
| | Hindu | 1.77(1.09, 2.87) | 0.021 | 1.78(1.09, 2.89) | 0.020 |
| Ethnicity | Brahmin | 1 | | 1 | |
| | Other (Janajati, Dalit, Madhesi) | 1.26(0.83, 1.90) | 0.278 | 2.11(1.39, 2.22) | <0.001 |
| Type of School | Public | 1 | | 1 | |
| | Private | 1.06(0.70, 1.60) | 0.789 | 1.23(0.80,1.87) | 0.344 |
| Education | Higher Secondary and above | 1 | | 1 | |
| | Secondary and below | 1.36(0.86,2.17) | 0.187 | 1.35(0.84, 1.16) | 0.209 |
| Mothers' education | Higher Secondary and above | 1 | | 1 | |
| | Secondary and below | 1.39(0.86, 2.26) | 0.179 | 2.40(1.46, 3.98) | <0.001 |
| Fathers' education | Higher Secondary and above | 1 | | 1 | |
| | Secondary and below | 0.99(0.63, 1.58) | 0.994 | 0.70(0.44, 1.13) | 0.146 |

at least one risk behavior whereas 14% were not having any NCDs risk factors. The prevalence of co-occurrence of NCDs risk factors was 41%. Older age participants, Hindus, non-Brahmins, and mothers' education (secondary and below) were positively associated with co-occurrence of NCDs risk factors.

According to the World Health Organization report 2014, physical inactivity is considered one of the fourth major global public health problems. Regular physical activity is a well-established protective measure for the treatment and prevention of common non-communicable diseases and their risk factors such as hypertension and diabetes [18]. Physical activity also plays an important role in maintaining healthy mental status and overall well-being of the individual ensuring quality of life [19]. This study showed 72.3% of the participants did not meet the WHO recommended physical activity for health. Almost 81% of adolescents aged between 11 to 17 years had inadequate physical activity globally [20]. A systematic review among South Asian adults revealed inconsistencies in the results, which varied between 18.5% - 88.4% [19]. Another study in urban districts of Nepal reported that 31% of the adolescents did not meet the WHO recommendation of physical activity [21]. The level of physical activity varies within and between the countries. In our study the higher prevalence of physical inactivity reported may be due to the lack of alternative transportation such as cycling lanes and schools' space for conducting extracurricular activities. Studies stated that the economic development of the countries and rapid changes in the means of transportation, urbanization, cultural and environmental factors influence the pattern of physical activity [22–24]. Age, gender, culture, socio-economic status, educational level, occupations are the major determinants in the status of physical activity [25, 26].

An unhealthy diet is linked with metabolic changes in the human body and thus increases the risk of non-communicable diseases [27]. This study showed that 41% of the school-going adolescents had an unhealthy dietary habits which is low compared to available evidence of 95% to 99% prevalence based on the studies at different settings in Nepal [5, 15, 28]. A study done in Kolkata, India showed that 80% of adolescents consumed an unhealthy diet [29]. Similarly, a study done in Kenya, Tanzania, and Brazil showed the highest proportion did not meet the recommended guidelines of WHO healthy diet [30–32]. But in contrast, a systematic review done on the adult population revealed 47% to 54% prevalence in unhealthy diets which is somehow consistent with this study [13]. The variation in the prevalence of unhealthy diets may be due to the socio-economic status, culture, environmental factors of the people. The eating patterns are mostly determined by the culture, region and economic status, and availability of the food [33, 34].

One of the major risk factors for non-communicable diseases is alcohol use. This study revealed that 15% prevalence of alcohol use in past 30 days which was similar to the finding of a study done in Kaski, Nepal [15]. In contrast, the prevalence was more than double the national report of GSHS 2015 Nepal [5].

The findings from this study showed 40.7% of the adolescents had co-occurrence of NCDs risk factors and 45.0% had one NCDs risk factor among school-going adolescents' students. This is similar to a study done in Pernambuco, Brazil which showed 47.3% of high school adolescents had two or more behavioral risk factors and only 37.7% had one non-communicable disease risk factor [35]. The findings are in contrast with the National School Health Survey done in ninth-grade students from both public and private schools of Brazil which reported 80.2% had any two or more co-occurrences of non-communicable diseases and behavioral risk factors [32]. A similar result to Brazil was reported by the study done in Pakistan which revealed 80% of the adolescents had two or more modifiable risk factors [36]. A study done in Kenya among the urban slum dwellers showed only 19.8% of the study participants had co-occurrence of non-communicable diseases risk factors and 52% of them had one risk factor

[16]. The wide range of differences between the study populations might be due to the small sample size and variation in study design. The result in this study shows inconsistencies but alarming situations of co-occurrence of non-communicable diseases. This needs to be addressed targeting the adolescents in prevention and controlling from adopting these harmful risk factors at an early stage of life through various interventional approaches.

In this study, the participants who followed Hindu and other than Brahmin/Chhetri ethnicity were more likely to have a co-occurrence of non-communicable diseases risk factors than non-Hindu religion and non-Brahmin ethnicity respectively. Similar results were shown by the study done in the semi-urban population of Pokhara where ethnicity other than Brahmin/Chhetri was found associated with risk factors of cardiovascular diseases [37]. Our study is also comparable to the other study done in the UK which shows caste/ethnicity were more likely to be involved in risk behavior [13]. In contrast, a study done in Dhulikhel did not find a significant association of religion and ethnicity with the risk factors for non-communicable diseases [38]. Our findings are likely because the majority of our sample belongs to the Hindu and other than Brahmin/Chhetri ethnicity group.

In this study, 46.3% of the mothers of school-going adolescents who had their schooling of secondary or below secondary level had co-occurrence of non-communicable diseases risk factors. A national school-based survey done in Brazilian adolescents shows association with lower education level of the mother with the co-occurrence of NCDs risk factors [32]. Another study in Nepal by using data from Nepal Demographic Health Survey also showed an association of risk factors with the lower educational level [6]. The findings of our study are in contrast with another study in the central west region of Brazil which reported an association with higher maternal education [39]. Evidence from the various literature showed the mother's education as an important influencing factor in the health outcome of the children.

This study had few limitations. First, there were chances of bias in the responses as the data were collected through a self-administered questionnaire. Second, assessment of physical activity, tobacco, alcohol, and diet was based on the response provided by the participants and not validated further by other methods. Third, the measurement of all four risk factors is self-reported thus may not free from under reporting due to social desirability bias. This study did not measure the chemical bio-maker to ascertain the use of alcohol and tobacco. As the study design was cross-sectional, the temporal association between co-occurrence of NCDs risk factors and socio-demographic variables could not be established. Longitudinal studies based on quantification of risk factors could overcome these limitations in the future.

## Conclusion

The co-occurrence of non-communicable disease risk factors was prevalent in two-fifths of adolescents. Almost seven in ten adolescents reported physical inactivity and two-fifths of adolescents had an unhealthy diet followed by alcohol consumption and tobacco use. Adolescent's ethnicity, religion, mother's education was found to be significantly and independently associated with the co-occurrence of non-communicable diseases risk factors.

## Supporting information

**S1 Table. Univariate analysis to determine factors associated with co-occurrence of NCDs risk factors.**
(DOCX)

**S1 Data.**
(SAV)

## Author Contributions

**Conceptualization:** Kalpana Tandon, Pranil Man Singh Pradhan.

**Data curation:** Kalpana Tandon, Nabin Adhikari, Bikram Adhikari, Pranil Man Singh Pradhan.

**Formal analysis:** Kalpana Tandon, Nabin Adhikari, Bikram Adhikari, Pranil Man Singh Pradhan.

**Investigation:** Kalpana Tandon.

**Methodology:** Kalpana Tandon, Nabin Adhikari, Bikram Adhikari, Pranil Man Singh Pradhan.

**Project administration:** Kalpana Tandon.

**Resources:** Kalpana Tandon.

**Software:** Kalpana Tandon, Nabin Adhikari, Bikram Adhikari.

**Supervision:** Kalpana Tandon, Nabin Adhikari.

**Validation:** Kalpana Tandon, Nabin Adhikari.

**Visualization:** Kalpana Tandon.

**Writing – original draft:** Kalpana Tandon, Nabin Adhikari, Bikram Adhikari, Pranil Man Singh Pradhan.

**Writing – review & editing:** Kalpana Tandon, Nabin Adhikari, Bikram Adhikari, Pranil Man Singh Pradhan.

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
