## [Decision Letter · Decision Letter 0]

11 Apr 2022

PONE-D-21-31588Co-occurrence of non-communicable disease risk factors and its determinants among school-going adolescents of Kathmandu Metropolitan CityPLOS ONE

Dear Dr. Pradhan,

Thank you for submitting your manuscript to PLOS ONE. After careful consideration, we feel that it has merit but does not fully meet PLOS ONE’s publication criteria as it currently stands. Therefore, we invite you to submit a revised version of the manuscript that addresses the points raised during the review process (see below).

We look forward to receiving your revised manuscript.

Kind regards,

Sahadat Hossain, MSPHI

Academic Editor

PLOS ONE

Journal Requirements:

3. We note you have included a table to which you do not refer in the text of your manuscript. Please ensure that you refer to Tables 2 and 4 in your text; if accepted, production will need this reference to link the reader to the Table.

4. Please include a copy of Table 12 which you refer to in your text on page 16.

Reviewers' comments:

Reviewer's Responses to Questions

**Comments to the Author**

1. Is the manuscript technically sound, and do the data support the conclusions?

Reviewer #1: Partly

Reviewer #2: Yes

Reviewer #3: No

Reviewer #4: Yes

Reviewer #5: No

2. Has the statistical analysis been performed appropriately and rigorously? 

Reviewer #1: No

Reviewer #2: Yes

Reviewer #3: No

Reviewer #4: Yes

Reviewer #5: No

3. Have the authors made all data underlying the findings in their manuscript fully available?

Reviewer #1: Yes

Reviewer #2: Yes

Reviewer #3: No

Reviewer #4: Yes

Reviewer #5: Yes

4. Is the manuscript presented in an intelligible fashion and written in standard English?

Reviewer #1: Yes

Reviewer #2: Yes

Reviewer #3: Yes

Reviewer #4: Yes

Reviewer #5: No

5. Review Comments to the Author

Reviewer #1: 1. The author points out in the study design that the age range of the subjected students is 13-19 years old, but 10-14 years old appears in Table 2, and 17-20 years old appears in line 234. The age definition and segmentation standards of the research objects are too confusing.

2. According to Table 1, Janajati has the highest share of Ethnicity，then why is Brahmin the definition of Ethnicity in Table 4?

3. In line 103, it is clear that since belonging to a public or private school will affect the content of this study, the proportion of students should be 1:1. This may bring great error to the research results of this paper.

4. In line 224, after the description of the data, the conclusion "Two-fifths of the school-going adolescents had co-occurrence of any two or more non-communicable disease risk factors" should be clearly stated so that the reader can understand the conclusions in the abstract.

Reviewer #2: Dear Editor,

This article aims to investigate the co-occurrence of non-communicable disease (NCDs) risk factors and its determinants among

school-going adolescents of Kathmandu Metropolitan City in Nepal. The authors focused on adolescents, who are of paramount importance in terms of identifying and modifying NCDs' risk factors, and thus this is an important paper to have in print. Here are some minor comments:

Abstract:

Please make sure that the conclusion completely follows the idea behind the study and reflects the main findings.

Introduction:

The main focus of the introduction must be on the NCDs among adolescents. The majority of the introduction is out of the funnel of the storyline of the introduction. I highly recommend to revise the section.

Discussion:

It is necessary to mention the authors' understanding of the article findings well.

Please review the discussion, check its storyline, and improve its coherency. It is not easy to follow in its current form.

Reviewer #3: Thank you for the opportunity to review the study entitled “Co-occurrence of non-communicable disease risk factors and its determinants among school-going adolescents of Kathmandu Metropolitan City.” The honorable authors have discussed the determine the prevalence and associated factors of co-occurrence of non-communicable disease risk factors among school-going adolescents of Kathmandu Metropolitan City. The paper is well written, and the Discussion section has described the observed results, although the study is subjected to a number of major limitations:

1. This study is cross-sectional, which only reveals the association. Also, the study has not any timing-based followed-up. So, the observed results only reveal the association, which is not sufficient. Based on current study, we could not conclude the accurate non-communicable disease risk factors.

2. The definition of the terms that are assessed in the method section is determined without any references, which provides additional biases for the way that authors have defined them.

3. Providing a figure regarding the results of study will help to convey the content of the study more sufficiently.

Reviewer #4: This school-based cross-sectional study by Kalpana Tandon, et al., 2022 determined the prevalence and associated factors of co-occurrence of non-communicable disease risk factors among school-going adolescents of Kathmandu Metropolitan City, Nepal.

The researchers used a validated structured questionnaire developed by the Global School Health

138 Survey 2015, World Health Organization and collected data from students, who were given 25-30 minutes to complete the form. The filled questionnaire was checked for completeness using variables:

Dependent (outcome) variable: co-occurrence of NCD risk factors

Independent variables: age, sex, 130 ethnicity types of school, level of 131 education, and parents’ education level

The independent variables (4) from all the 1108 participants who filled the questionnaire, having fulfilled the inclusion criteria, performed descriptive analysis and presented parametric numerical variables with mean and standard deviation and categorical variables with percentage and frequency. The Clopper-Pearson method was used to determine the confidence interval of prevalence. The multivariate multinomial logistic regression was applied to determine independent factors associated with co-occurrence of NCDs risk factors after adjusting for age, gender, 165 ethnicity, religion, education, type of school, and parental education.

The authors made all data underlying the findings fully available. The data was tested for representativeness, analyzed using descriptive and inferential statistics which were rigorous and appropriate.

Results obtained revealed that prevalence of physical inactivity, unhealthy diet, harmful use of alcohol and tobacco among school-going adolescents were 72.3% (95%CI: 69.6-74.9), 41.1%

44 (95%CI: 38.2-44.0), 14.8% (95%CI: 12.8-17.0) and 7.8% (95%CI:6.3-9.5) respectively.

Discussions of the results were robust, citing similar studies conducted both within and outside Nepal.

Conclusions are in line with the findings

Writing quality and clarity: Satisfactory

Other observations:

1. Limitations of the study: The authors did well to mention the limitations of the study but they fell short of suggesting how these limitations should be addressed by future studies going forward

2. No inclusion/exclusion criteria were stated

References: The manuscript employed the use of Harvard style referencing but requires editing to correct some errors noticed e.g., Listing of references: Shouldn’t this be in alphabetical order? Shouldn’t the journal name be italics? Shouldn’t the list of authors that are more than 5 be reflected as et al?

I suggest the authors should revise Harvard referencing style and make necessary corrections.

Reviewer #5: I appreciate the opportunity to review this article titled, "Co-occurrence of non-communicable disease risk factors and its determinants among school-going adolescents of Kathmandu Metropolitan City". Results of a cross-sectional survey of 1108 adolescents in grades 9-12 are presented in this manuscript. Due to the high prevalence of risk factors for NCDs in low-income countries, the paper's topic is relevant and timely. In general, three concerns should be addressed: first, the background section needs a clear explanation for the study; second, the methodology section is lacking and requires more information about the analysis process (e.g., how were the variables selected for the multivariate model); and third, the results sections need better organization guided by a clear framework. Due to the limited scope of the findings presented in the paper, it does not add more value to the existing evidence.

6. PLOS authors have the option to publish the peer review history of their article (what does this mean?). If published, this will include your full peer review and any attached files.

Reviewer #1: **Yes: **Weina Liu

Reviewer #2: No

Reviewer #3: No

Reviewer #4: **Yes: **Haruna Ismaila ADAMU, MBBS; MPH; PhD; MACE

Reviewer #5: No

---

## [Author Response · Author response to Decision Letter 0]

4 May 2022

Overall comment:

1) We note you have included a table to which you do not refer in the text of your manuscript. Please ensure that you refer to Tables 2 and 4 in your text; if accepted, production will need this reference to link the reader to the Table.

Tables are correctly indicated in the text 

2) Please include a copy of Table 12 which you refer to in your text on page 16.

Table 12 was typographical error which is corrected.

Reviewer 1

1) The author points out in the study design that the age range of the subjected students is 13-19 years old, but 10-14 years old appears in Table 2, and 17-20 years old appears in line 234. The age definition and segmentation standards of the research objects are too confusing.

The difference in age category label between methods, Table 2 and line 234 is because of typographical error which is corrected after checking the dataset which is available in the open data registry.

2) According to Table 1, Janajati has the highest share of Ethnicity then why is Brahmin the definition of Ethnicity in Table 4?

The Janajati, Dalit and Madhesi groups considered as ethnic minorities and they share nearly similar characteristics (like drinking, non-vegetarian diet etc) but does not match with the Brahmin Chhetri ethnic groups. Hence Brahmin/Chhetri group is kept as separate group.

3) In line 103, it is clear that since belonging to a public or private school will affect the content of this study, the proportion of students should be 1:1. This may bring great error to the research results of this paper.

The descriptive analysis is added stratified by type of school (public/private). In multivariate regression, the type of school is also adjusted.

4) In line 224, after the description of the data, the conclusion "Two-fifths of the school-going adolescents had co-occurrence of any two or more non-communicable disease risk factors" should be clearly stated so that the reader can understand the conclusions in the abstract.

It is clarified in the abstract and in result section.

Reviewer 2

1) Abstract: Please make sure that the conclusion completely follows the idea behind the study and reflects the main findings.

Conclusion part in abstract is edited so that it follows the idea behind the study and main findings.

2) Introduction: The main focus of the introduction must be on the NCDs among adolescents. The majority of the introduction is out of the funnel of the storyline of the introduction. I highly recommend to revise the section.

Relevant Content are added to the introduction part

3) Discussion: It is necessary to mention the authors' understanding of the article findings well. Please review the discussion, check its storyline, and improve its coherency. It is not easy to follow in its current form.

Discussion has been revised as per suggestion.

Reviewer 3

1. This study is cross-sectional, which only reveals the association. Also, the study has not any timing-based followed-up. So, the observed results only reveal the association, which is not sufficient. Based on current study, we could not conclude the accurate non-communicable disease risk factors.

I agree with you that cross-sectional study is not appropriate to determine risk factors. In this study we have determined factors associated with co-occurrences of different established NDCs risk factors, not the factors associated with NCDs. We have used STEPS survey to find the whether the school going adolescents have four different NCDs risk factors or not. 

2. The definition of the terms that are assessed in the method section is determined without any references, which provides additional biases for the way that authors have defined them.

References are now added to the definition of terms in method section

3. Providing a figure regarding the results of study will help to convey the content of the study more sufficiently.

Figures are added to the manuscript

Reviewer 4

1. Limitations of the study: The authors did well to mention the limitations of the study but they fell short of suggesting how these limitations should be addressed by future studies going forward

We have revised the limitations as per your suggestion.

2. No inclusion/exclusion criteria were stated

Inclusion and exclusion criteria are added in the participant recruitment section of manuscript

3. References: The manuscript employed the use of Harvard style referencing but requires editing to correct some errors noticed e.g., listing of references: Shouldn’t this be in alphabetical order? Shouldn’t the journal name be italics? Shouldn’t the list of authors that are more than 5 be reflected as et al? I suggest the authors should revise Harvard referencing style and make necessary corrections.

We used Vancouver referencing style specific to PLOS One as referencing guideline. 

Reviewer 5

1) The background section needs a clear explanation for the study

Background is modified for clear explanation

2) The methodology section is lacking and requires more information about the analysis process (e.g., how were the variables selected for the multivariate model)

It is added in the manuscript. In the model we added socio-demographic variables- age, gender, ethnicity, religion, education, type of school, and parental education (mother’s education, and father’s education) with p-value less than 0.50 in univariate analysis (S 1 Table of supplementary file).

3) The results sections need better organization guided by a clear framework. Due to the limited scope of the findings presented in the paper, it does not add more value to the existing evidence.

The results are now modified.

---

## [Decision Letter · Decision Letter 1]

18 Jul 2022

Co-occurrence of non-communicable disease risk factors and its determinants among school-going adolescents of Kathmandu Metropolitan City

PONE-D-21-31588R1

Dear Dr. Pradhan,

We’re pleased to inform you that your manuscript has been judged scientifically suitable for publication and will be formally accepted for publication once it meets all outstanding technical requirements.

Kind regards,

Jianhong Zhou

Staff Editor

PLOS ONE

Additional Editor Comments (optional):

Reviewers' comments:

Reviewer's Responses to Questions

**Comments to the Author**

1. If the authors have adequately addressed your comments raised in a previous round of review and you feel that this manuscript is now acceptable for publication, you may indicate that here to bypass the “Comments to the Author” section, enter your conflict of interest statement in the “Confidential to Editor” section, and submit your "Accept" recommendation.

Reviewer #1: All comments have been addressed

Reviewer #2: All comments have been addressed

Reviewer #3: All comments have been addressed

Reviewer #4: All comments have been addressed

2. Is the manuscript technically sound, and do the data support the conclusions?

Reviewer #1: Yes

Reviewer #2: Partly

Reviewer #3: Yes

Reviewer #4: Yes

3. Has the statistical analysis been performed appropriately and rigorously? 

Reviewer #1: Yes

Reviewer #2: Yes

Reviewer #3: Yes

Reviewer #4: Yes

4. Have the authors made all data underlying the findings in their manuscript fully available?

Reviewer #1: Yes

Reviewer #2: Yes

Reviewer #3: Yes

Reviewer #4: Yes

5. Is the manuscript presented in an intelligible fashion and written in standard English?

Reviewer #1: No

Reviewer #2: Yes

Reviewer #3: Yes

Reviewer #4: Yes

6. Review Comments to the Author

Reviewer #1: Minor concerns:

1. Typing errors should be avoided, such as in the abstract and table 1, etc.

2. The first person is not suitable for use in the abstract of this manuscript.

3. The standardization and aesthetics of the tables should be improved.

4. In the discussion section, the statement “Our findings are likely because the majority of our sample belongs to the Hindu and other than Brahmin/Chhetri ethnicity group” is unreasonable. It is recommended to focus on the characteristics of different ethnic groups.

Reviewer #2: (No Response)

Reviewer #3: Dear Authors,

I read the revised version of the manuscript.

The authors have addressed the comments and included additional data for supporting the results. The introduction section has provided the required information for emphasizing the implications of this study. Also, additional figures help authors to visualize data more effectively.

Reviewer #4: My observations in the previous review have been adequately addressed. I have no additional comments to make at this time.

7. PLOS authors have the option to publish the peer review history of their article (what does this mean?). If published, this will include your full peer review and any attached files.

Reviewer #1: No

Reviewer #2: No

Reviewer #3: No

Reviewer #4: **Yes: **Haruna Ismaila Adamu, MBBS; MPH; PhD

---

## [Editor Report · Acceptance letter]

29 Jul 2022

PONE-D-21-31588R1 

Co-occurrence of non-communicable disease risk factors and its determinants among school-going adolescents of Kathmandu Metropolitan City 

Dear Dr. Pradhan:

I'm pleased to inform you that your manuscript has been deemed suitable for publication in PLOS ONE. Congratulations! Your manuscript is now with our production department. 

Kind regards, 

on behalf of

Jianhong Zhou 

Staff Editor

PLOS ONE